# KnapFormer: An Online Load Balancer for Efficient Diffusion Transformers Training

## Abstract

We present KnapFormer, an efficient and versatile framework to combine workload balancing and sequence parallelism in distributed training of Diffusion Transformers (DiT). KnapFormer builds on the insight that strong synergy exists between sequence parallelism and the need to address the significant token imbalance across ranks. This imbalance arises from variable-length text inputs and varying visual token counts in mixed-resolution and image-video joint training. KnapFormer redistributes tokens by first gathering sequence length metadata across all ranks in a balancing group and solving a global knapsack problem. The solver aims to minimize the variances of total workload per-GPU, while accounting for the effect of sequence parallelism. By integrating DeepSpeed-Ulysees-based sequence parallelism in the load-balancing decision process and utilizing a simple semi-empirical workload model, KnapFormers achieves minimal communication overhead and less than $1\%$ workload discrepancy in real-world training workloads with sequence length varying from a few hundred to tens of thousands. It eliminates straggler effects and achieves $2\times$ to $3\times$ speedup when training state-of-the-art diffusion models like FLUX on mixed-resolution and image-video joint data corpora. We attach the source code of our KnapFormer implementation in the supplementary materials.

## 1 Introduction

Large-scale generative vision-language models, such as Diffusion Transformers (DiT) (Peebles & Xie, 2023), are typically trained on multimodal inputs that combine both text and image (or video) content. In these settings, each training sample generally contains a text prompt of variable length and a visual input of varying spatial resolution. After tokenization, these inputs produce multimodal token sequences whose lengths can vary significantly across a batch. This variability arises primarily due to (1) differences in the length of text prompts and (2) mixed-resolution visual inputs and joint image-video training strategies (Chai et al., 2022; Gu et al., 2023; Gao et al., 2025), which yields variable numbers of visual tokens even after standard aspect ratio bucketing techniques (Anlatan / NovelAI, 2022).

In distributed training scenarios—such as data-parallel pretraining on multi-GPU clusters—this token-length heterogeneity creates a substantial workload imbalance across devices. As a result, some GPUs are overloaded with longer token sequences while others are underutilized, leading to:

- Straggler effects (Dean & Ghemawat, 2008), where the slowest GPU dictates the synchronization point for the entire system
- Suboptimal GPU utilization and degraded training throughput
- Increased memory pressure on overloaded devices, potentially leading to out-of-memory (OOM) errors

Achieving balanced token-level workloads across GPUs is crucial for high-performance training. However, existing strategies are often insufficient in practice. Static bucketing (Anlatan / NovelAI, 2022) requires manual planning for workload balancing. Fixed-length padding schemes can achieve workload balancing also with manual balancing. But it wastes compute in padded tokens and fails to address fine-grained imbalance, especially in heterogeneous setups involving multi-resolution

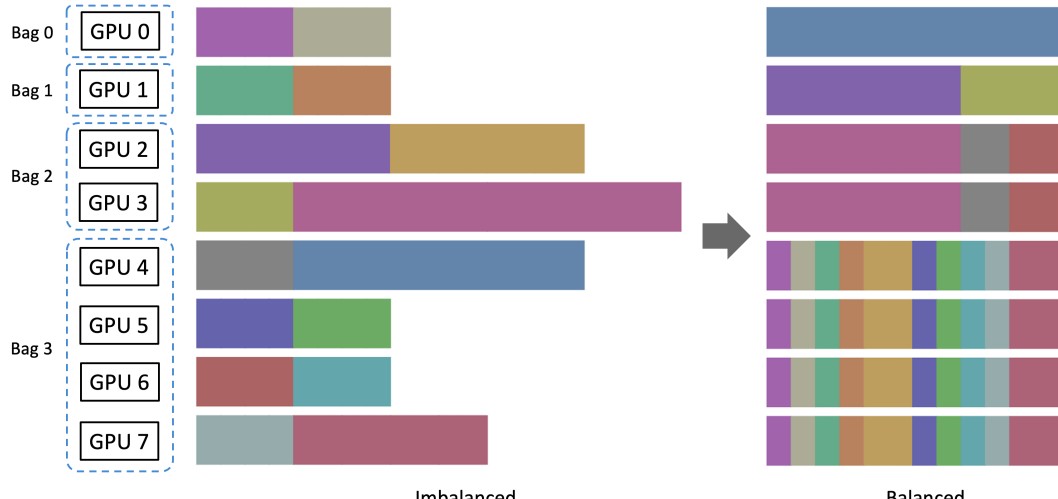

Figure 1: In KnapFormer, GPUs are logically grouped into compute bags, each responsible for processing a subset of sequences. A global multi-knapsack problem is solved on the CPU to assign sequences to bags in a way that minimizes workload imbalance. The resulting mapping is then executed on the GPU via a single all-to-all collective. For bags containing multiple GPUs, assigned sequences are evenly chunked across devices to enable parallel processing within each bag.

inputs, joint image-video corpora, and variable-length text prompts. Moreover, manually crafted heuristics for workload balancing, while labor-intensive, struggle when the sequences change dynamically in training.

We introduce KnapFormer, a general-purpose, efficient, and reversible sequence-chunk-level workload balancing algorithm designed for distributed multimodal pretraining to address these challenges. KnapFormer performs online workload balancing, effectively addressing the need to balance dynamic workloads. It minimizes per-GPU compute variance by redistributing token sequences across devices based on their lengths, using a global assignment strategy inspired by the classic multi-knapsack problem. Notably, such redistribution is performed using a single all-to-all communication, which incurs minimal communication overhead.

KnapFormer also supports compute groups that span multiple GPUs—referred to as *compute bags*—which enable sequences to be partitioned across devices within a bag. This design naturally supports sequence parallelism and is compatible with efficient distributed attention algorithms such as Ulyssys attention (Jacobs et al., 2023). KnapFormer includes a reverse mapping—also implemented via an all-to-all collective—that restores the original sequence order to support downstream tasks such as collation or loss computation.

Our contributions are summarized as follows:

- We propose KnapFormer, a fast sequence-chunk-level balancing algorithm, with native support for sequence parallelism, that significantly reduces workload variance across GPUs, eliminating stragglers and boosting training throughput in the case of highly heterogeneous inputs.

- We provide an efficient and scalable KnapFormer implementation, designed as a plug-in module into the existing code, using PyTorch collective communication primitives and requiring only a single all-to-all operation per redistribution.

- Our method supports fully reversible redistribution for transparent loss computation and is compatible with auxiliary features like per-token positions, which ease integration into pre-existing training systems.

Experimental results show that KnapFormer leads to more balanced GPU workloads and improves training efficiency by a large margin in large-scale diffusion transformer training.

## 2 RELATED WORK

Balancing token-level workloads across GPUs is a central challenge in large-scale multimodal training. Existing solutions span a spectrum of approaches: from dynamic routing in Mixture-of-Experts (MoE) models to fine-grained intra-layer partitioning in sequence and tensor parallelism. While effective in some scenarios, these techniques often require careful manual tuning, especially in the presence of heterogeneous data sources (e.g., mixed-resolution images, joint image-video corpora, and variable-length text). In contrast, KnapFormer provides a carefree, data-agnostic balancing mechanism that adapts automatically to input variability without heavily modifying model architecture or manually adjusting parallelism strategy per dataset.

**Mixture-of-Experts (MoE).** Our method shares similarities with Mixture-of-Experts (MoE) models (Shazeer et al., 2017; Huo et al., 2025; Yang et al., 2025; Dai et al., 2024; Dat; Jiang et al., 2024), which introduce sparsity by routing each token to a subset of expert MLPs, typically distributed across GPUs. In expert-parallel MoE implementations (Lepikhin et al., 2020; Gale et al., 2023), this per-token routing results in significant inter-device communication during every transformer block.

In contrast, KnapFormer performs coarse-grained, chunk-level routing of token sequences, and only twice per iteration in optimal scenarios—once before the forward pass and once after the backward pass. This minimizes communication overhead outside the transformer blocks. Moreover, while MoE models primarily aim to increase model capacity, our focus is on balancing compute loads across GPUs for standard dense models, improving throughput without requiring specialized routing architectures.

**Sequence and Tensor Parallelism.** KnapFormer is also closely related to model parallelism strategies such as sequence parallelism(Li et al., 2021) and tensor parallelism(Shoeybi et al., 2019). In our setup, when multiple GPUs are grouped into a compute bag, sequences are chunked across GPUs similar to sequence parallelism for the feedforward layers. For attention layers, we adopt Ulyssys attention (Jacobs et al., 2023), which distributes attention heads across devices, akin to tensor parallelism. One concurrent work on integrating load-balancing with sequence parallelism is MAGI-attention (Zewei & Yunpeng, 2025), which utilizes RingAttention (Liu et al., 2023) for sequence parallelism. It requires a specific attention implementation and token redistribution in each attention operation to achieve workload balancing. Knapformer, instead, can support arbitrary attention implementation and does not require additional communication over DeepSpeed Ulyees.

We believe the key distinction of KnapFormer lies in usability and adaptability. Traditional model parallelism strategies often require careful reconfiguration of sharding strategies to accommodate datasets with differing input modalities, resolutions, or token length distributions. This manual tuning is especially burdensome in multimodal or mixed-domain training pipelines. In contrast, KnapFormer provides automatic, data-driven token redistribution, enabling consistent and balanced workload assignment across GPUs—regardless of input heterogeneity—without heavily modifying core model internals.

## 3 METHOD

We cover details of our method in this section. On the high level, our method consists of an gamma-corrected workload model (Sec. 3.1), a compact syntax specifying compute topology (Sec. 3.2), a greedy load balancer (Sec. 3.3), and integration with Ulysys attention (Sec. 3.4). We also introduce the APIs of KnapFormer package (Sec. 3.5).

### 3.1 LOAD MODELING

Processing a token sequence of length $l$ with embedding dimension $d$ in a standard transformer block incurs a compute cost of:

$$w = 24ld^2 + 4l^2d, \tag{1}$$

as noted in prior work (Casson, 2023). The first term, $24ld^2$, accounts for linear projections in the feedforward and attention modules (including query, key, value, and output projections), while the second term, $4l^2d$, corresponds to the attention softmax operation, which scales quadratically with the sequence length.

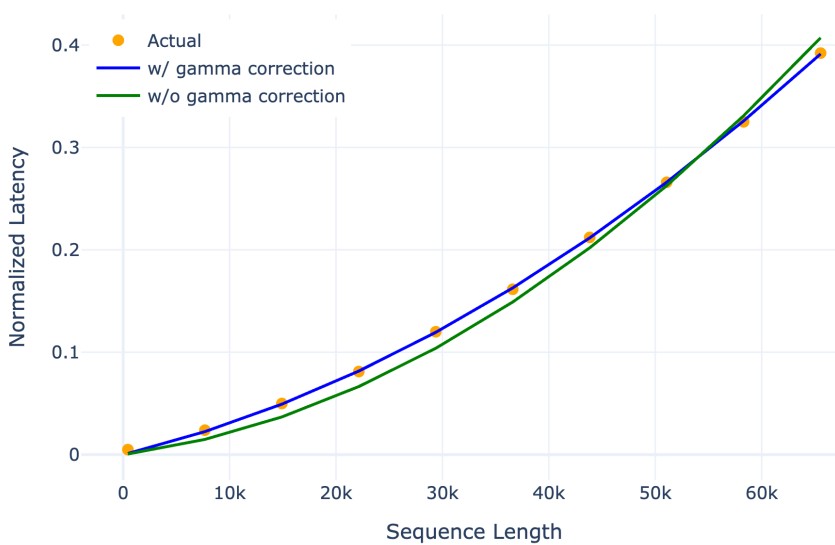

Figure 2: Latency prediction based on the FLOPs (Eq. 1) is a bit off from the empirical observations, while our gamma-corrected formula (Eq. 2) provides more accurate latency estimates. The estimated $\gamma = $ is 0.385 for FLUX model on H100 GPUs.

In practice, our goal is to minimize latency $t$ with workload balancing. When the system is compute-bound, latency scales approximately linearly with the number of FLOPs, i.e., $t \approx kw$, where $k$ is a hardware-dependent constant. However, we observe empirically that the attention term tends to be more memory-bandwidth bound and does not scale linearly with FLOPs count. To better match real-world measurements, we introduce a correction factor $\gamma$ that is specific to a GPU architecture and refine the latency model as:

$$t = k \cdot (24ld^2 + \gamma \cdot 4l^2 d), \tag{2}$$

where $\gamma$ is obtained by fitting the model to measured $(l, t)$ pairs. As shown in Figure 2, this correction improves latency estimation accuracy across different sequence lengths.

## 3.2 COMPUTE TOPOLOGY SPECIFICATION

To support efficient distributed training with variable-length sequences, we define a 2D logical GPU topology that groups devices into compute bags. Each compute bag consists of one or more GPUs assigned to jointly process a set of input sequences. This design allows long or compute-intensive sequences to span multiple GPUs, while shorter sequences can be assigned to smaller bags—enabling both flexibility and efficiency in workload placement.

The topology is configured using a concise string format: **g1n2+g2n1+g4n1**, which specifies two 1-GPU bags (g1n2), one 2-GPU bag (g2n1), and one 4-GPU bag (g4n1). Each term gGnN denotes $N$ bags of size $G$. This defines a sharding unit consisting of $GN$ GPUs, to which token sequences are redistributed for balanced processing. The entire GPU cluster is then partitioned into repeated replicas of this unit, ensuring uniform scaling and full hardware utilization.

Despite operating on homogeneous clusters, this logical compute bag abstraction enables adaptive and fine-grained sequence placement. Samples are dynamically assigned to bags based on estimated token workload, avoiding excessive padding or overcommitment of resources. This approach naturally accommodates the heterogeneity of real-world data (e.g., mixed-length text, variable-resolution images, or video sequences) within a unified scheduling and redistribution framework.

**Special Case: Intra-Node Parallelism.** A particularly effective topology is to group all 8 GPUs within a node into a single compute bag, specified as g8n4 for a 4-node setup. In this configuration, each sequence is split into 8 contiguous chunks and distributed across GPUs within the same node. This layout results in an almost perfect load balancing no matter how heterogenous the data

sources are. It leverages high-bandwidth inter-GPU communication via NVIDIA NVLink (nvi) in the Ulysss attention part of each transformer block, reducing latency and improving throughput compared to cross-node message passing.

## 3.3 GREEDY LOAD BALANCER

In each compute group (or replica), let there be $M$ compute bags and a total of $G$ GPUs. Our goal is to assign $N$ variable-length sequences to these bags and GPUs in a way that minimizes per-GPU workload imbalance. The algorithm proceeds in three passes:

**First Pass — Assign Sequences to Bags (No Data Movement).**

- Compute the estimated workload $w_i$ for each sequence $i$, forming a list of $(i, w_i)$ pairs.
- Compute the target per-GPU workload as $\frac{\sum_{i=1}^{N} w_i}{G}$.
- Determine the capacity $c_j$ of each bag $j$ as the product of the number of GPUs in the bag and the per-GPU target workload.
- Sort the sequence list in descending order of $w_i$.
- For each sequence in sorted order:
    - Identify bags where the remaining capacity is sufficient for $w_i$.
    - Among those, select the bag with the lowest current *occupancy*, defined as the ratio of total assigned workload to bag capacity.
    - Assign the sequence to the selected bag and update its occupancy.

**Second Pass — Assign Sequence Chunks to GPUs (No Data Movement).**

- For each sequence:
    - Retrieve the number of GPUs in its assigned bag.
    - Evenly partition the sequence into that many contiguous chunks, assigning one chunk per GPU in the bag.

**Third Pass — Route Chunks to Target GPUs (With Data Movement).**

At this stage, we have a complete mapping: $\text{chunk}_k, \text{source-GPU}_k, \text{target-GPU} k_{k=1}^{K}$. To perform the actual redistribution, we invoke a single GPU-side `all-to-all` collective, which transfers all chunks to their target GPUs. This operation is implemented using differentiable PyTorch distributed primitives, enabling end-to-end gradient flow without detaching computation graphs.

## 3.4 INTEGRATION WITH ULYSSES ATTENTION

When a sequence is assigned to a compute bag with more than one GPU, it is divided into partial sequences and distributed across the GPUs in the bag. However, standard attention mechanisms require access to the full sequence context. To address this, we integrate the Ulysses attention algorithm (Jacobs et al., 2023), which enables distributed attention computation with minimal communication overhead.

Specifically, Ulysses performs an efficient transformation between two representations:

- **(Partial sequences, full heads):** Each GPU holds a local chunk of the sequence but all attention heads.
- **(Full sequences, partial heads):** Each GPU holds the full sequence but only a subset of heads.

One important property of Ulysees-based distributed attention is that we can easily achieve an equal workload split of attention computation in a sequence parallelism group by having each worker operate on the same number of attention heads. This allows our simple per-sequence workload model to remain effective in planning with sequence parallelism on any distribution of sequence

lengths and attention masks. Distributed attention approaches based on online Softmax, such as RingAttention Liu et al. (2023), do not have this property and require complex workload models and token redistribution per Transformer block Zewei & Yunpeng (2025) to achieve load-balancing.

We perform a single intra-bag `all-to-all` operation to switch from the first to the second layout. This layout enables direct use of the optimized FlashAttention kernels (Dao et al., 2022; Dao, 2024) on each GPU, which operate on full sequences with reduced memory overhead. After computing attention, a second `all-to-all` restores the original layout.

This integration allows us to achieve sequence parallelism in attention layers with minimal changes to the model code, while maintaining both efficiency and scalability.

**Use the KnapFormer API outside transformer blocks**

```
from knapformer import SequenceBalancer

# g1n2+g2n1+g4n1 means: 2 1-GPU Bags, 1 2-GPU Bag, 1 4-GPU Bag
load_balancer = SequenceBalancer("g1n2+g2n1+g4n1")

# this_gpu_seq_lens: [l1, l2, ..., ln]
# this_gpu_packed_seqs: Tensor of shape (l1+l2+...+ln, d_model)
# this_gpu_packed_features: list of Tensors;
#                     each of shape (l1+l2+...+ln, d_feature)
load_balancer.plan_routing(this_gpu_seq_lens, d_model)
bala_chunk_lens, bala_seqs, bala_features = load_balancer.route(
    this_gpu_packed_seqs, this_gpu_packed_features
)
for _ in range(num_layers):
    bala_seqs = transformer_block(
        bala_seqs, bala_chunk_lens, bala_features, load_balancer
    )
seq_lens, seq_out = load_balancer.reverse_route(bala_seqs)
```

**Use the KnapFormer API inside the attention block**

```
def apply_attn(
    packed_q: torch.Tensor,
    packed_k: torch.Tensor,
    packed_v: torch.Tensor,
    load_balancer: SequenceBalancer,
):
    # packed_q, packed_k, packed_v: (l'1+l'2+...+l'n, h, d)

    # For single-GPU bag, do nothing;
    # For bags with >1 GPUs, do Ulysses:
    #   (partial sequences, full heads)->(full sequences, partial heads)
    seq_lens, packed_q, packed_k, packed_v = load_balancer.pre_attn(
        packed_q, packed_k, packed_v
    )

    # Users can leverage Flash-Attention varlen kernels
    #   or customize their own attention masks using seq_lens
    packed_x = scaled_dot_product_attention(
        packed_q, packed_k, packed_v, seq_lens
    )

    # For single-GPU bag, do nothing;
    # For bags with >1 GPUs, do Ulysses:
    #   (full sequences, partial heads)->(partial sequences, full heads)
    seq_lens, packed_x = load_balancer.post_attn(packed_x)

    return packed_x
```

## 3.5 API DESIGN

To make sure our load balancer can be easily plugged into existing transformer models, we design the function interfaces in the following way. We first show the use of the load balancer in transformer training on a high level, then demonstrate how it interacts with the attention operator inside each transformer block.

## 4 EXPERIMENTS

To showcase our load balancer's improvement made to training efficiency, we build a training simulator for text-to-{image, video} diffusion training. It consists of the synthetic data generation (Sec. 4.1) and a simple training loop shown below. We primarily focus on the efficiency of the model forward/backward part in a practical FSDP2-based distributed training setup (pyt). We also apply activation checkpointing (Beaumont et al., 2019) to each transformer block for simplicity.[1]

**Fake training loop**

```
1  def simulate_training (model_config, data_config, balancer_config):
2      model = create_transformer(model_config)
3      data_simulator = create_data_simulator(data_config)
4      load_balancer = create_load_balancer(balancer_config)
5
6      for seq_lens, seq_ins in data_simulator.next_batch():
7          # seq_lens: list with values [l1, l2, ..., ln]
8          # seq_ins: tensor of shape [l1+l2+...+ln, d_model]
9          seq_outs = model(seq_lens, seq_ins, load_balancer)
10         seq_outs.mean().backward()   # backward with dummy loss
11         for p in model.parameters(): # set gradients to None
12             p.grad = None
```

**Group 0: data loader sharding on GPU 0-31**

| | Data code | # GPU | GPU indices | Batch size per GPU | Batch size per 32 GPU | Avg. text tokens per datum | Avg. visual tokens per datum | Avg. tokens per GPU | Avg. tokens per 32 GPU |
|---|---|---|---|---|---|---|---|---|---|
| **Images** | g8b4i256f1s0 | 8 | 0-7 | 4 | 32 | 196 | 256 | 1808 | 14464 |
| | g2b5i512f1s0 | 2 | 8-9 | 5 | 10 | 196 | 1024 | 6100 | 12200 |
| | g2b5i1024f1s0 | 2 | 10-11 | 5 | 10 | 196 | 4096 | 21460 | 42920 |
| | g4b1i2048f1s0 | 4 | 12-15 | 1 | 4 | 196 | 16384 | 16580 | 66320 |
| **Keyframes** | g1b10i256f4s0 | 1 | 16 | 10 | 10 | 196 | 1024 | 12200 | 12200 |
| | g3b1i512f4s0 | 3 | 17-19 | 1 | 3 | 196 | 4096 | 4292 | 12876 |
| **Videos** | g8b2i256f85s1 | 8 | 20-27 | 2 | 16 | 196 | 6400 | 13192 | 105536 |
| | g4b1i512f85s1 | 4 | 28-31 | 1 | 4 | 196 | 25600 | 103184 | 412736 |

**Group 1: repeat group 0 on GPU 32-63**

⋮

**Group N: repeat group 0 on GPU 32N-32N+31**

Figure 3: **Example 2D heterogeneous data configurations for joint image and video pretraining.** Similar to HSDP designed for model sharding, we have both replica and sharding groups for the set of different data loaders as well. In this example, each sharding group consists of 32 GPUs for hosting heterogeneous data sources including pairs of (text, single image), (text, video keyframes) and (text, video clip) data.

### 4.1 SYNTHETIC DATA STREAMS FOR TEXT-TO-{IMAGE, VIDEO} DIFFUSION TRAINING

Our 2D heterogeneous dataloader is designed to efficiently utilize large-scale GPU clusters by sharding different data streams across subsets of GPUs and then replicating the same configuration across multiple groups.

As shown in Fig. 3, each data stream is defined by a compact data code of the form **g{G}b{B}i{R}f{F}s{S}**, where **G** specifies the number of GPUs the stream is sharded across, **B** is the per-GPU batch size, **R** indicates the spatial resolution (e.g., i256 for a 256×256 image), **F**

---

[1]In practice, one should apply activation checkpointing selectively to a subset of the transformer blocks and operators for maximum efficiency, depending on the actual memory budget under a specific training configuration of data, model, optimizer and balancer.

is the number of frames (temporal extent), and **S** is the smoothness parameter, which determines whether temporal compression is applied.

The number of tokens per sample is computed by dividing the resolution and frame count by the VAE's spatial and temporal compression rates, and a random multiplier is applied to the spatial token count to simulate varying aspect ratios. In this 2D layout, the sharding axis introduces heterogeneity across GPUs by assigning different streams (e.g., image, keyframe, video) to different ranks, while the replication axis ensures that this mixed configuration is repeated identically across multiple GPU groups.

Factoring the DiT patch sizes, we set the VAE spatial compression rate to 16; temporal compression rate is set to 3.4 (17 pixel-space frames are compressed into 5 latent frames; it's only applicable to video clips but not sparse keyframes). The number of text tokens is randomly chosen between 0 and 392 for each sample independently, averaging 196 text tokens per sample. To account for aspect ratio bucketing, we multiply the number of visual tokens by a factor randomly chosen in $[0.96, 1.04]$ for all the samples in a batch; this corresponds to an aspect ratio bucketing strategy where aspect ratios fall inside $[1/4, 4]$.

## 4.2 RESULTS

### Low-resolution image pretraining
**Data codes:** g32b32i256f1s0

|              | WIR ↓ | FBL ↓ | TPS ↑    | HFU ↑   |
|--------------|-------|-------|----------|---------|
| w/o Balancer | 1.22  | 3.13s | 147.63K  | 24.72%  |
| Balancer **g1n32** | 1.00  | 2.95s | 156.48K  | 26.20%  |
| Balancer **g2n16** | 1.00  | 3.30s | 139.77k  | 23.40%  |
| Balancer **g4n8**  | 1.00  | 3.51s | 131.63k  | 22.04%  |
| Balancer **g8n4**  | 1.00  | 4.30s | 107.28k  | 17.96%  |

### Mixed-resolution image pretraining
**Data codes:** g16b4i256f1s0, g4b5i512f1s0, g4b5i1024f1s0, g8b1i2048f1s0

|              | WIR ↓ | FBL ↓ | TPS ↑    | HFU ↑   |
|--------------|-------|-------|----------|---------|
| w/o Balancer | 17.14 | 4.64s | 58.58K   | 14.54%  |
| Balancer **g1n32** | 3.92  | 4.62s | 58.95k   | 14.63%  |
| Balancer **g2n16** | 1.27  | 2.92s | 93.08k   | 23.10%  |
| Balancer **g4n8**  | 1.01  | 2.62s | 103.74k  | 25.74%  |
| Balancer **g8n4**  | 1.00  | 2.52s | 108.06K  | 26.81%  |

### Joint image and video pretraining
**Data codes:** g8b4i256f1s0, g2b5i512f1s0, g2b5i1024f1s0, g4b1i2048f1s0
g1b10i256f4s0, g3b1i512f4s0, g8b2i256f85s1, g4b1i512f85s1

|              | WIR ↓ | FBL ↓ | TPS ↑    | HFU ↑   |
|--------------|-------|-------|----------|---------|
| w/o Balancer | 28.11 | 8.05s | 45.88K   | 12.69%  |
| Balancer **g1n32** | 4.61  | 8.11s | 45.56k   | 12.60%  |
| Balancer **g2n16** | 1.62  | 4.71s | 78.40k   | 21.68%  |
| Balancer **g4n8**  | 1.00  | 3.55s | 104.07k  | 28.78%  |
| Balancer **g8n4**  | 1.00  | 3.44s | 107.54K  | 29.74%  |

Table 1: **FLUX DiT.** Comparison of training efficiency for text-to-{image,video} diffusion models with and without load balancer. We benchmark on 32 H100 GPUs using our training simulator where the load balancing is performed on these 32 GPUs. The estimated $\gamma = 0.49$ on H100 GPUs in the workload equation. FLUX uses 19 DoubleStream and 38 SingleStream transformer blocks with adaLN layers (Peebles & Xie, 2023), totaling 12B parameters; in each block, we have d_model=3072, d_head=128. *Note that we report HFU as opposed to MFU, to better align with the real use case where activation checkpointing is used. One can further dial up the HFU by increasing the total batch sizes at each step.*

We test our load balancer on 32 H100 GPUs in three practical pretraining scenarios: 1) low-resolution image pretraining; 2) mixed-resolution image pretraining; 3) joint image and video pretraining. We choose the data mixing ratios based on common large-scale diffusion training recipes. We use the following metrics to measure our balancer's performance:

- **Workload Imbalance Ratio (WIR)**: we define the WIR as the ratio between the maximum and minimum per-GPU total workloads.
- **Forward-Backward Latency (FBL)**: we only consider model forward and backward time, excluding the effects of different optimizers' varying efficiency.
- **Tokens Per Sec (TPS)**: we aggregate the throughput over all GPUs participating in the training simulation.
- **Hardware-FLOPs Utilization (HFU)**: suppose the model forward FLOPs is $m$; we consider the forward FLOPs ($m$), backward FLOPs ($2m$) and activation recomputation FLOPs ($m$), which sums to $4m$ FLOPs in total.

We report the main results in Tab. 1. For different training scenarios, we sweep over 4 homogeneous compute topologies from g1n32, g2n16, g4n8 and g8n4. In the low-resolution image training setup, the workload is relatively balanced due to the more homogeneous data configurations where the sequence length variations mainly come from varying-length text captions and image aspect ratio bucketing. The balancer g1n32 leads to a decent 5% speed-up. However, other balancer setups slow down the training due to the extra communication involved. Our method shows much bigger speedups when the data sources become highly heterogeneous in the case of mixed-resolution image training and joint image-video training. Without any optimization, the HFU can be very slow, causing huge compute wastes. By integrating the simple g8n4 balancer configuration, we manage to double the training throughput. Moreover, we find that g8n4 seems to work the best compared with other homogeneous compute topologies like g1n32, g2n16, g4n8. However, there might be better non-homogeneous compute topologies; we leave this to future work.

## 5 CONCLUSION

In this work, we introduced KnapFormer, an online sequence-chunk-level workload balancing algorithm that maximizes training throughput for transformer models under highly heterogeneous data distributions. By dynamically redistributing sequences across GPUs based on their compute cost, KnapFormer eliminates stragglers and improves hardware utilization—without requiring heavy manual tuning. Our method simplifies system design in two key ways: (1) it removes the need for complex load balancing in the data loader, and (2) it offers a lightweight alternative to manually configured sequence parallelism. Thanks to its modular design, KnapFormer can be readily integrated into existing training pipelines with minimal modifications. Beyond training, our load balancer can also be applied during inference to improve system throughput in real-world deployments, especially when user inputs vary significantly in length, aspect ratio, or resolution.

**Limitations and Future Work.**  While KnapFormer shows strong performance in balancing heterogeneous workloads, the compute topology (i.e., GPU bag specification) currently is static and pre-specified before training. In this work, we rely on simple patterns such as g8nN (i.e., grouping 8 GPUs per node into a bag). An important direction for future research is to develop methods that automatically infer optimal compute topologies—especially non-uniform or heterogeneous bag configurations—based on the input data distribution and hardware topology. It would also be interesting to improve upon the current greedy load balancing strategy. Finally, we assume homogeneous attention masks inside each datum (either bidirectional full attention or causal attention). There remains space for improvement for more irregular attention mask patterns used by unified multi-modal generative models (Deng et al., 2025; Zhou et al., 2024).

**LLMs usage:** In preparing this manuscript, we used large language models (LLMs) as general-purpose writing assistants for grammar corrections, rephrasing, and clarity/concision edits. All LLM-suggested edits were reviewed and verified by the authors, who take full responsibility for the final manuscript.

**Reproducibility statement.** We have included sufficient implementation details in the method section of the main paper to reproduce the experimental results. Moreover, we also include a copy of the source code in the supplementary materials.

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
