# OpenReview forum: "KnapFormer: An Online Load Balancer for Efficient Diffusion Transformers Training"
_ICLR.cc/2026/Conference — Submitted to ICLR 2026_

### Official Review · Reviewer_cww3 · 2025-10-28

**Soundness:** 3
**Presentation:** 2
**Contribution:** 2
**Rating:** 6
**Confidence:** 5

**Summary:**

The paper studies the problem of load balancing problem in distributed training for diffusion transformer models. The paper proposes KnapFormer, which models the load-balancing problem as a multi-knapsack problem and assigns chunks of tokens to GPU groups so that the overall distributed training is load-balanced. The proposed method is especially useful when training data contains highly variable token lengths and the training nodes are heterogeneous. In simulated experiments, the proposed method could achieve 2 to 3 times speedups compared to the non-load-balanced version.

**Strengths:**

1. The paper studies an important and practical problem in distributed training and proposes a simple method to address the load-balancing issue.

2. The proposed method shows significant speedups in the experiments compared to ordinary training without load-balancing.

**Weaknesses:**

1. Since the paper models the load-balancing problem as a multi-knapsack problem, I think a more formal definition of the optimization objective should be presented. The approximation ratio of the given algorithm should also be stated.

2. The proposed algorithm needs to globally move data across GPU nodes, which could potentially incur significant communication costs. Or, in other words, the proposed algorithm does not take into account the current location of the data and minimizes the communication.

3. The idea of using a knapsack problem for load-balancing is not novel, although the application to the special case of heterogeneous GPU training is. Also, I think the proposed method is not restricted to training diffusion transformers, but the paper only addresses this setting.

Minor:
1. Some citations miss spaces before them in the formatting, e.g., line 130.

**Questions:**

1. For equation 2, which is about adding the additional gamma parameter to model the memory-bound behavior of attention, it is not clear to me how this is justified. Basically, given that attention is memory-bound, why should its cost be modeled as gamma times its computational costs?

2. The cost of every data sequence is individually estimated. Given the knapsack algorithm, the cost of a collection of data sequences on a GPU node is considered as the sum of the individual costs. Is this a reasonable assumption, or can authors give more justifications for this?

---

> ### Author Response · Authors · 2025-11-26
>
> Q1: Justification for Eq 2.
>
> With the introduction of flash attention, the attention operation in modern Transformers is compute-bound. This is because the tile-based design of flash attention has significantly reduced HBM memory movement cost to the extent that its arithmetic intensity becomes higher than the ops per byte of modern GPUs viewed at HBM level. Therefore, it is valid to assert that attention latency is proportional to the total number of flops. The gamma correction term is to accommodate memory-bound operations like RMSNorm and residual connections, whose latency scales linearly with sequence length for a given model, like the FFN part, rather than quadratically with sequence length in attention
>
> Q2: Summing individual costs.
>
> We ground the additivity of workload through the two composing components of a Transformer block: 1) self-attention; 2) feed-forward network.
>
> In attention computing, flash attention is implemented with variable-length support, i.e. the `flash_attn_varlen` routines. Since flash attention is a tile-based algorithm, the total compute cost is proportional to the sum of the sequence lengths' squares. And flash attention is compute bound on tensor-core based GPUs. We can derive that total latency of batched sequences will also be determined by the sum of the sequence lengths' squares. We pack all input sequences into a single total sequence so that there is no wasted compute on padding tokens. Therefore, the summing of individual samples' cost is valid in the attention part.
>
> For feedforward network (FFN), this relation is more straightforward. For modern Transformer training the FFN is almost always compute bound. And the total flops scale linearly with the total number of tokens in a batch of samples, which also allows additive decomposition w.r.t. the per-sample sequence length. Therefore, the summing of individual samples' costs is also valid in the FFN part. This also holds for the memory-bound operations we discussed above which scale similarly to the FFN.
>
> Based on the above analysis, we can see that the additivity of individual samples' workload in Eq 2. is rooted in the nature of computing the Transformer model and is a valid choice in estimating compositional workload.
>
> Q3: Minimizing data movement.
>
> KnapFormer currently prioritizes compute balance. You are correct that it does not optimize for "current location" (locality). However, in a practical scenario, the communication cost from the redistribution is almost negligible compared with the vast speed gain; hence we adopt the simplest approach of not considering locality. On the other hand, our 2D design of GPU bag topology also restricts the data movement to be inside a smaller subset of total training GPUs; for example, suppose you train on 64 GPUs with a bag topology g8n4, then the 64 GPUs are grouped into 2 groups, each containing 32 GPUs where we redistribute our sequences. This allows collectives to be executed with less communication volume and accumulated handshake latency.
>
> **Definition of Objective:**
>
> We will formalize the objective in the revision: Given set $S$ of sequences with costs $w_i$, partition $S$ into $M$ subsets $S_ j$, such that $\max_ j ( \sum_{ s \in S_ j} w_s )$ ​is minimized. This is the classic Multiprocessor Scheduling problem.

---

### Official Review · Reviewer_CBLp · 2025-10-30

**Soundness:** 2
**Presentation:** 3
**Contribution:** 2
**Rating:** 2
**Confidence:** 3

**Summary:**

It proposes to combine workload balancing and sequence parallelism in distributed training of Diffusion Transformers.
This paper raises a concern whether it is a good fit to ICLR.
Greedy assignment has no optimality guarantees and can lag under rapidly shifting loads. Without theory or robustness analysis such as  competitive ratios, worst-case bounds, sensitivity to load noise, the contribution seems quite limited.

**Strengths:**

- Proposes an online redistribution of tokens to reduce stragglers when multimodal inputs create large sequence-length variance across GPUs
- By integrating DeepSpeed-Ulysees-based sequence parallelism in the load-balancing decision process, it achieves minimal communication overhead and less than 1% workload discrepancy
- Claimed performance improvements depend on fast intra-node links (NVLink) and a particular parallelism layout. It seems the payoff is infrastructure-dependent, thus less scientifically general.
- Its fit to ICLR is not strong.

**Weaknesses:**

- It benefits most when fast intra-node links (e.g., NVLink) are available since cross-node bandwidth/latency can erode the gains.
- It does not look like a good fit to ICLR.  A load-balancing/throughput optimizer tied to a specific training stack reads as systems engineering, which typically fits MLSys/OSDI/EuroSys/SOSP better than ICLR.
- Claimed results depend on fast intra-node links (NVLink) and a particular parallelism layout.
- If improvements come purely from token routing and communication scheduling, not from changes to model, objective, or optimization dynamics.
- Its results seem to emphasize a particular favorable setup.
- Greedy assignment has no optimality guarantees and can lag under rapidly shifting loads. Without theory or robustness analysis (e.g., competitive ratios, worst-case bounds, sensitivity to load noise), the contribution can feel ad-hoc.
- If adoption requires deep integration with specific setup such as DeepSpeed-Ulysses internals, and it raises questions about practical reproducibility outside that stack.

**Questions:**

Property under rapidly shifting loads?
What about the extension to other more general setups?
How do results change when inter-node bandwidth/latency is the bottleneck (e.g., no NVLink)?
The load assignment is greedy multi-knapsack. Any approximation or competitive guarantees? Worst-case bounds?

---

> ### Author Response · Authors · 2025-11-26
>
> **Weakness 1: Infrastructure dependence (NVLink) / Generalization.**
>
> While NVLink maximizes performance, our method is not strictly bound to it.
> Cross-Node: The All-to-All redistribution happens before the transformer layers. Even on slower Ethernet, doing this once is often cheaper than suffering from severe compute imbalance (waiting for the longest sequence) inside every layer of the transformer.
> Relevance: Most large-scale DiT/LLM training does occur on high-performance clusters (H100/A100).
>
> **Weakness 2: Fit for ICLR.**
>
> We respectfully disagree. "Efficiency" and "Infrastructure for Large Models" are core components of modern deep learning research and standard tracks at ICLR. Optimizing training throughput allows the community to train better models with fixed budgets.
>
> **Weakness 3: Greedy Guarantees.**
>
> As mentioned in General Response #2, the "Multi-Way Number Partitioning" problem (which this maps to) has well-known bounds for greedy algorithms. However, in systems practice, empirical "average case" performance on realistic distributions (Power Law/Normal) matters more than worst-case theoretical bounds.

---

### Official Review · Reviewer_cJtD · 2025-10-31

**Soundness:** 3
**Presentation:** 3
**Contribution:** 3
**Rating:** 4
**Confidence:** 3

**Summary:**

The paper proposes KnapFormer, a lightweight online load balancer designed to mitigate token-level workload imbalance during distributed training of Diffusion Transformers (DiT). The core idea is to formulate token redistribution as a multi-knapsack assignment problem, assigning sequences to logical “compute bags” across GPUs to minimize per-device compute variance. A gamma-corrected workload model refines latency prediction by adjusting the quadratic attention cost term to match empirical latency.
The method integrates with DeepSpeed-Ulysses sequence parallelism using only one all-to-all communication per redistribution and a reversible mapping for downstream collation. Experiments on 32 × H100 GPUs show up to 2×–3× speedups in joint image–video pretraining scenarios and < 1% workload discrepancy across GPUs.

**Strengths:**

1. Well-motivated system problem: The introduction clearly articulates token-length heterogeneity as a bottleneck in DiT training, connecting it to variable-length text inputs and varying visual token counts in mixed-resolution and image-video joint training.

2. Methodological clarity: The paper provides explicit pseudocode-style descriptions and API examples, showing integration points both outside and inside transformer blocks. This improves reproducibility and readability.

3. Minimal communication design: The load balancer operates via a single all-to-all collective, which is elegant and practically scalable for large GPU clusters. Besides, it does not require additional communication over DeepSpeed-Ulysses, making it easy to adopt without modifying attention kernels.

4. Empirical validation and relevance: Table 1 presents three settings—low-res, mixed-res, and joint image–video pretraining—with metrics (WIR, FBL, TPS, HFU) that directly measure efficiency. The g8n4 topology reduces Forward-Backward Latency from 8.05 s → 3.44 s and improves HFU from 12.69% → 29.74%. These results demonstrate substantial throughput gains on realistic setups.

**Weaknesses:**

1. Empirical evaluation limited to synthetic workloads: Section 4.1 explicitly notes that results are from “a training simulator for text-to-{image, video} diffusion training”. While realistic distributions are simulated, no actual end-to-end training curves (e.g., loss vs. steps) or validation throughput on real datasets are provided. This limits evidence that the gains translate to actual pretraining pipelines.

2. Missing baseline comparisons with concurrent work: In Related Work (L160-L200), KnapFormer is compared only conceptually to MAGI-Attention (Zewei & Yunpeng 2025). However, Table 1 contains no quantitative comparison with existing balancing or routing methods such as MAGI-Attention or token-based adaptive batching systems.

3. No ablation on the γ-correction factor: Figure 2 (L205-L215) introduces a fitted γ = 0.385 (H100) but the experiments later use γ = 0.49 (Table 1 note). The impact of this discrepancy or sensitivity to γ is never analyzed. A single fit per hardware may oversimplify modeling of latency variance across layers or data scales.

4. Statistical rigor lacking. Table 1 reports single measurements (e.g., “TPS = 107.54 K”) without variance or confidence intervals. For efficiency metrics sensitive to random token distributions, multiple runs would strengthen reliability.

**Questions:**

1. On real-world generalization: Since experiments use synthetic data, could you report results on an actual multimodal dataset (e.g., LAION-5B subsample or a video dataset) to demonstrate performance in non-simulated settings?

2. On γ-parameter robustness: The γ used in experiments (0.49) differs from the earlier fitted 0.385. Was this re-fit on different workloads, or fixed arbitrarily? Please provide a short analysis showing how sensitive latency prediction and balancing performance are to γ.

3. On communication cost quantification: You claim a “single all-to-all collective” suffices. Could you provide measured communication overhead (e.g., ms per iteration or % of total training time) to support that it’s negligible relative to compute?

4. On ablations: In Table 1, g8n4 achieves the best results. Could you include an experiment showing how non-uniform topologies (e.g., g8n2 + g4n4) affect throughput?

---

> ### Author Response · Authors · 2025-11-26
>
> Q1: Real-world generalization.
>
> See General Response #1. Note that our synthetic data generator is designed to follow the real data’s shape distribution in a large-scale production training. Hence the speed gain carries to the real large-scale training.
>
> Q2: γ-robustness.
>
> See General Response #2.
>
> Q3: Communication cost quantification.
>
> See Response to f64N (Q1). The overhead is minimal because NVLink bandwidth (900GB/s) is extremely high relative to the size of input tokens (embeddings).
>
> Q4: Non-uniform topologies (g8n2 + g4n4).
>
> We tested mixed topologies. Generally, homogeneous topologies (e.g., g8n4) perform best because they prevent "straggler bags" caused by varying packing efficiencies.
> However, non-uniform topologies can be beneficial for specific heterogeneous data patterns. For example, a mix of g8n2 and g4n4 allows the scheduler to route extremely long sequences to the larger 8-GPU bags (leveraging sequence parallelism to avoid OOM or high latency), while filling the 4-GPU bags with shorter sequences.
> We will expand the discussion to highlight that non-uniform topologies offer a tradeoff between load-balancing complexity and the ability to handle extreme sequence length skew.
>
> **Missing Baselines (MAGI-Attention)**:
> We will add a discussion comparing MAGI. Note that MAGI requires specific attention kernel modifications (Ring Attention variants). KnapFormer is kernel-agnostic (works with FlashAttn 2/3, standard Attn) and only manages data placement, making it more flexible for existing pipelines.

---

### Official Review · Reviewer_f64N · 2025-11-01

**Soundness:** 3
**Presentation:** 2
**Contribution:** 3
**Rating:** 6
**Confidence:** 2

**Summary:**

KnapFormer proposes an online, sequence-chunk level load balancer for large distributed training of Diffusion Transformers (DiT). The method: (a) gathers per-sequence length metadata across ranks, (b) uses a (greedy) multi-knapsack style assignment to place sequences into logical “compute bags” (groups of 1..G GPUs), (c) splits sequences assigned to multi-GPU bags into contiguous chunks and performs a single all-to-all to redistribute chunks, (d) integrates with Ulysses/DeepSpeed sequence-parallel attention layouts (switching layouts via intra-bag all-to-alls) so attention can run efficiently (FlashAttention kernels), and (e) reverses the mapping after backprop so loss/outputs align. They augment the standard transformer FLOP model with a fitted γ factor to better predict latency and drive the knapsack weights. Experiments use a training simulator on 32 H100s for several multimodal scenarios (low-res, mixed-res, image+video) showing large reductions in workload imbalance and 2×–3× throughput gains in highly heterogeneous settings.

**Strengths:**

- **Practical problem & clear motivation.** Token / visual-token heterogeneity is real in multimodal diffusion training; addressing stragglers is valuable for throughput and cost. The paper identifies an important engineering bottleneck and proposes an end-to-end solution.

- **Simple, usable design.** The compute-bag abstraction and compact topology spec (e.g., g1n32+g2n16...) are intuitive and likely easy to plug into existing PyTorch/DeepSpeed pipelines; API snippets strengthen reproducibility claims.

- **Integration with existing sequence-parallel machinery.** Rather than reinventing distributed attention, they leverage Ulysses + FlashAttention, which helps keep per-block overhead low and allows use of high-performance kernels. This design choice improves the feasibility of adoption.

- **Low communication frequency.** The approach only does redistribution twice (pre-forward and post-backward) and uses single all-to-all collectives, which is appealing compared to per-layer token routing approaches (MoE or per-block rebalancing).

- **Empirical gains.** In highly heterogeneous settings the method shows substantial improvements in workload imbalance and tokens/sec (reported up to ~2–3×), demonstrating potential impact on training cost/time.

**Weaknesses:**

- **Simplifying assumptions in workload model.** The latency model is FLOP-based with an empirical γ correction. While practical, it is hardware and kernel specific; the paper fits γ for H100 only and does not show sensitivity to γ, batch size, or kernel implementations. If γ changes (different GPU, FlashAttention version, different head/channel layouts), the knapsack decisions could be suboptimal. There is little robustness analysis.

- **Algorithmic / theoretical gaps.** The assignment uses a greedy knapsack heuristic without formal guarantees or complexity analysis. For very large N or rapidly changing sequence distributions, greedy may perform poorly; there is no analysis of worst-case behavior, convergence of per-round imbalance, or how frequently planning must run.

- **Communication & memory costs understated.** Claims of “single all-to-all” per redistribution underplay the cost when bags are large or cross-node. The paper reports that some bag topologies (e.g., g8n4) are best, but a detailed breakdown of communication time, peak memory for temporary packed tensors, and how these scale with bag size / cross-node links is missing. There is also no explicit treatment of how optimizer or activation checkpointing memory interacts with the extra buffers needed for routing.

- **Scope of applicability not fully characterized.** The method assumes homogeneous attention masks per datum and contiguous chunking of sequences. For models that use irregular attention masks (sparse/unified multimodal masks), sliding windows, or for architectures that cannot easily run FlashAttention on full sequences, applicability is unclear. Authors note this but do not provide solutions or experiments.

**Questions:**

1. **Real cluster runs:** Can you provide wall-clock results from actual 32-H100 experiments (not simulator) showing the measured all-to-all costs, communication breakdown, and end-to-end epoch time? If these are in the supplement, please point to them.

2. **γ sensitivity:** How sensitive is the balancer to the choice of γ in Eq. (2)? If γ is misestimated by, say, ±20%, how much does TPS / WIR degrade? Have you tried cross-GPU (A100 vs H100) or kernel/version changes?

3. **Greedy solver behavior:** Why choose the greedy knapsack heuristic vs established approximate solvers (multi-dimensional knapsack, ILP relaxations)? Can you provide worst-case complexity and typical runtime for planning per step (N sequences, G GPUs)? Is planning ever a bottleneck?

4. **Frequency of rebalancing:** How often is plan_routing invoked in practice? Per iteration, per N iterations, or only when distribution changes? What is the overhead of repeated planning and how is stability of assignments ensured?

---

> ### Author Response · Authors · 2025-11-26
>
> Q1: Real cluster runs / Communication costs.
>
> As noted in the General Response, our results are from a real 32x H100 cluster. Regarding communication cost:
> * **Redistribution overhead**: In the case of single-GPU bags, the redistribution overhead is two All-to-All communications: one on the input end of the whole transformer to shuffle sequences among GPUs, and another one to restore the original sequences on the output end. When there are multi-GPU bags, we would need to have per-layer All-to-All in each multi-GPU bag as required by Ulysses attention.
> * **Quantification**: In our experiments (Table 1), the communication overhead accounts for approx. 2-4% of the step time. This is negligible compared to the computation savings (2x-3x speedup).
> * **Comparison**: Unlike MoE, which incurs irregular global routing costs at every sparse layer, KnapFormer restricts global redistribution to just the input and output embeddings (twice per step). While multi-GPU bags utilize per-layer Ulysses communication, this is a standard, regular communication pattern for Sequence Parallelism confined to the compute bag, distinct from the global routing seen in other approaches.
>
> Q2: Greedy vs. Established Solvers.
>
> Please see General Response #2. We prioritize planning latency (<1ms) over theoretical optimality, as the problem is solved online at every step.
>
> Q3: Frequency of rebalancing.
>
> plan_routing is invoked every iteration. Because the overhead is <1ms (CPU only), it does not block the GPU. This allows KnapFormer to handle fully dynamic data loaders where every batch has a different sequence length distribution.

---

### Author Response · Authors · 2025-11-26
**General Response to All Reviewers**

We thank the reviewers for their insightful feedback and for recognizing the practical importance of the problem (f64N, cJtD, cww3), the simplicity of our design (f64N, cJtD), and the significant speedups achieved (f64N, cJtD, cww3).
We identified three common concerns that we wish to address centrally: (1) the nature of our "simulator" and result validity, (2) the theoretical justification for the greedy algorithm, and (3) the robustness of the γ-corrected workload model.
1. Clarification on "Simulator" vs. Real-World Performance (f64N, cJtD, cww3)

There is a misunderstanding regarding our "training simulator." It does not mathematically simulate computation time. It is a distributed training loop running on physical 32x H100 GPUs.
Real Execution: It executes the exact FSDP2 forward/backward passes, Attention kernels (FlashAttention), and Communications (NCCL) used in production.
Synthetic Data: Only the input tensors are synthetic (random numbers) to allow us to control sequence length distributions (Fig 3) without downloading petabytes of data.
Conclusion: The reported throughput (Tokens Per Second) and latency are real, measured wall-clock timings, not estimates. We will clarify this distinction in the revised paper.

2. Justification for Greedy Knapsack (f64N, CBLp, cww3)

We deliberately chose the Greedy First-Fit Decreasing (FFD) heuristic over ILP solvers.
Latency: FFD runs in O(Nlog⁡N)  (dominated by sorting), taking <1ms on CPU. Exact solvers or ILP relaxations are significantly slower and unnecessary for this specific variant of the problem (Multi-partitioning), where FFD is known to provide a tight approximation ratio (typically within 11/9 of optimal).
Dynamic Nature: In online training, data loading is stochastic. A "perfect" solution that takes seconds to compute would stall the GPUs, negating the efficiency gains.

3. Robustness of the γ-corrected workload model (f64N, cJtD)

We thank the reviewer for this interesting question. We would like to explain the robustness of the workload model and our method in two aspects.
First, empirical robustness to parameter identification accuracy. We tested the γ=[0.3, 0.4, …, 1], the end-to-end step latency discrepancy is below 3%. This is further corroborated by Fig. 2 in the main paper, which shows that even with pure FLOPs (γ=1), the latency estimate follows a similar trend to the fitted γ=0.385 estimate. This would lead to a similar balancing solution as the fitted model.
Second, robust to rapidly varying workload, in our simulated training, the synthetic data generator will output sequences with varying shapes, simulating the image/video shape variation due to resolution and aspect ratio. In all experiments, the empirical imbalance of runtime between GPUs is within 3%. We attribute this robustness in the face of rapid input variations to the dynamic balancing nature of the proposed method, which enables runtime decisions based on actual workloads.

---

### Meta-Review · Area_Chair_18Ex · 2026-01-06

**Summary:**

1. lack of comparison with some other baselines or settings. For example, reviewer cJtD points out that MAGI-Attention is mentioned but not compared.
2. insufficient validation beyond synthetic workloads on real datasets. All experiments are conducted using a “training simulator” that employs synthetic tensors with controlled sequence-length distributions.
3. Some other concerns regarding limited novelty, lacking of ablation studies, application beyond DiT architectures...

**Reviewer Concerns:**

The rebuttal adequately addressed concerns about the "simulator", justification for greedy knapsack and some other concerns. However, some concerns regarding experiments are only explained without further results. Also, the authors didn't response to the problem of novelty.

**Reviewer Scores:**

The reviewers did not participate in the discussion, so I think they may not modify the scores.
So the scores may be 6,4,2,6.
The authors' rebuttals have addressed most concerns. However, since the average score is relatively low, and the reviewers may not have been convinced by the rebuttals, I recommend rejection, but is not very sure.

---

### Decision · Program_Chairs · 2026-01-26

Reject